# Synthesis and Characterization of Hybrid Metal Zeolitic Imidazolate Framework Membrane for Efficient H_2_/CO_2_ Gas Separation

**DOI:** 10.3390/ma13215009

**Published:** 2020-11-06

**Authors:** Po-Hsueh Chang, Yuan-Tse Lee, Cheng-Hsiung Peng

**Affiliations:** 1Department and Institute of Chemical and Materials Engineering, Minghsin University of Science and Technology, Xinxing Road, Hsinchu 30401, Taiwan; pohsueh.chang@gmail.com; 2Department of Materials Science and Engineering, National Chiao Tung University, 1001 University Road, Hsinchu 300, Taiwan; xu3m06yk6@msn.com

**Keywords:** zeolitic imidazolate framework, ZIF-8-67 membranes, gas separation, separation factor, permeance

## Abstract

In this paper, we propose mixed metal ions in the node of the zeolitic imidazolate framework (ZIF) structure. The hybrid metal ZIF is formed for the gas separation of hydrogen and carbon dioxide. In the first stage, the nanoparticles were prepared as a coating on a substrate, and acting as secondary growing nuclei. The hybrid metal ZIF structures were characterized by X-ray diffractometry (XRD) and Fourier transform infrared spectroscopy (FTIR). N_2_ adsorption–desorption isotherms determined surface area, and scanning electron microscopy (SEM) was used to observe the microstructure and surface morphology. The hybrid metal ZIF-8-67 powder had the largest surface area (1260.40 m^2^ g^−1^), and the nanoparticles (100 nm) could be fully dense-coated on the substrate to benefit the subsequent membrane growth. In the second stage, we prepared the hybrid metal ZIF-8-67 membrane on the pre-seeding substrate with mixed metal nanoparticles of cobalt and zinc, by the microwave hydrothermal method. Cobalt ions were identified in the tetrahedral coordination through UV–Vis, and the membrane structure and morphology were determined by XRD and SEM. Finally, a gas permeation analyzer (GPA) was used to determine the gas separation performance of the hybrid metal ZIF-8-67 membrane. We successfully introduced zinc ions and cobalt ions into the ZIF structure, where cobalt had a strong interaction with CO_2_. Therefore, GPA analysis showed an excellent H_2_/CO_2_ separation factor due to lower CO_2_ permeability. The CO_2_ permeance was ~0.65 × 10^−8^ mol m^−2^ s^−1^ Pa^−1^, and the separation factors for H_2_/CO_2_ and H_2_/N_2_ were 9.2 and 2.9, respectively. Our results demonstrate that the hybrid metal ZIF-8-67 membrane has a superior H_2_/CO_2_ separation factor, which can be attributed to its very high specific surface area and structure. Based on the above, hybrid metal ZIF-8-67 membranes are expected to be applied in hydrogen or carbon dioxide gas separation and purification.

## 1. Introduction 

Zeolitic imidazolate frameworks (ZIFs) are members of a new class of hybrid organic–inorganic materials, called metal–organic frameworks (MOFs), which have relatively regular pores and significant advantages: in a higher specific surface area, ranging from micropores to mesopores; designable structure characteristics, including the skeleton, pore shape, pore size, and surface chemical properties [1]. ZIFs have ordered porous structures with hybrid frameworks, which consist of inorganic metal ions or metal clusters coordinated with organic imidazolate ligands [2,3]. Both the versatility and designability of the organic ligands, and the directionality of the metal ion, offer not only conventional adsorptive properties, such as storage, separation, and catalysis, but also other physical/chemical functions integrated into the frameworks. In addition, they have a highly flexible structure, including pore size and surface properties that can be rationally designed. The material’s structural properties have relatively high thermal stability and physical–chemical stability, which makes them appear as excellent candidates for superior-performance gas separation or purification membranes [2]. The structural composition of ZIFs consists of metal atoms (such as Zn and Co) regularly connected with ditopic imidazolate (Im), or functionalized Im, and N atoms to form neutral frameworks [4].

To date, of the practical applications of gas separation that have been undertaken [5], and with energy efficiency as the primary consideration, membrane-based separation has advantages compared with pressure swing adsorption and cryogenic distillation. Although several MOF membranes have been synthesized by different methods [6], ZIF membranes still have better gas separation performance [7]; however, their structure and function control remains a challenge. Bux et al. successfully prepared a zeolitic imidazole framework-8 (ZIF-8) membrane with molecular sieving properties using microwave-assisted synthesis. In addition, the results showed that the ZIF-8 membrane achieved a balance between permeability and selectivity [8]. Recently, Pan et al. prepared a membrane with excellent gas separation performance, combining the ZIF-8 membrane on the surface of hollow yttrium-stabilized zirconia (YSZ) fiber by an environmentally friendly seeded growth method [9]. On the other hand, ZIF-67 is one of the most studied materials, and its microporous framework structure is formed by cobalt cations linked 2-methylimidazolate anions [10]. ZIF-67 has remarkable chemical stability and a large surface area (>1000 m^2^/g), and is widely used for the selective separation of CO_2_, O_2_, N_2_, and CH_4_ [11]. However, due to the microporous nature of ZIF-67, which has a small pore size (1.16~0.34 nm), the adsorption properties are obstructed by the slow diffusion kinetics [12,13]. This phenomenon severely limits the application of ZIF-67 in separation, adsorption, and catalysis. In particular, the problem may become more serious when the size of the adsorbed molecule is similar to or larger than the micropore diameter.

So far, most developed ZIFs have been based on a single metal. Many intrinsic properties of ZIFs, such as thermal stability, chemical resistance, and structural density, depend on the metal composition. Therefore, to effectively and stably configure the composition of mixed metal ZIFs will be an effective method to regulate their essential properties. Each metal can form separated clusters, resulting in a mixture of discrete ZIFs. These add to the difficulty of synthesizing mixed metal ZIFs. The synthesis of mixed metal ZIFs generally uses linkers with free reactive functional groups. However, the functional groups can coordinate with other metals not only pre-synthesis, but also post-synthesis. Few attempts have involved the gas selective embeddedness of the different metals, and included a mixture of metal in the framework’s nodes. Cobalt was selected for this study because of its excellent interaction properties with CO_2_, making the membrane useful for gas separation applications [14,15]. It has been reported that the ligand 2-methylimidazole (mIM) reacting with Zn or Co salts can be used to prepare ZIF-8 or ZIF-67, respectively.

Previously, the approach commonly used was to design novel materials, chemically modified using functional groups. Some strategies have recently been proposed to tune the properties of MOFs, and control framework porosity and functionality, by mixed-linker synthesis [16,17]. Some ZIFs of Zn^2+^ and Co^2+^ with mIM (ZIF-8 and ZIF-67) have also been developed, but few studies have been conducted on mixed metal ZIFs synthesis, and studied their properties. Botas et al. [18] reported that they synthesized a framework of Zn_4_O(bdc)_3_, and then replaced Zn^2+^ with Co^2+^ through ion replacement. The results showed that the adsorption capacity of H_2_, CO_2_, and CH_4_ under high pressure was better than samples without Co^2+^. Zhang et al. [19] prepared mixed-matrix membranes (MMMs) by embedding Ni^2+^ replaced ZIF-8 to form Zn/Ni based ZIF-8, and then making a polyether block amide polymer, which was applied to CO_2_/N_2_ mixed gas separation. The results showed that the ratio of 10% Zn/Ni based ZIF-8 had the best gas separation capacity, with a CO_2_ flux of 282 Barrer, and selectivity of 42.7, for CO_2_/N_2_, respectively. Fan et al. [20] used imidazole, polyimides, and ZIF-8 to prepare membranes with a high separation performance through an exquisite synthesis method. A gas separation analysis displayed the selectivity of H_2_/CH_4_, which increased with the increase of the ZIF-8 ratio, resulting in a selectivity of 318.3, while H_2_ permeability was approximately 72.3 Barrer.

However, the direct synthesis strategy may provide a simpler process for preparing a novel hybrid metal ZIF-8-67 membrane, with highly stable and excellent gas separation of H_2_/CO_2_. This strategy was adopted in this work and can hopefully approach the ideal membrane performance. 

We report the synthesis of hybrid metalZIF-8-67 membranes, c-oriented by the microwave hydrothermal method. This study aims to compare the difference in H_2_/CO_2_ gas separation efficiency between single metals (ZIF-8 and ZIF-67) and hybrid metals (ZIF-8-67). In addition, the optimal cobalt content, characterization, and theoretical mechanism of c-oriented crystal growth, dependent on the preparation route, is discussed, which was further observed by scanning electron microscopy (SEM) and X-ray diffraction (XRD) analysis.

## 2. Experimental Methods

### 2.1. Synthesis of ZIF-8-67 Nanocrystals

Synthesis of ZIF-8-67 nanocrystals was undertaken as follows. A 0.5:0.5:16:16 Zn:Co:ligand:TEA of molar ratio 0.366 g Zn(NO_3_)_2_·6H_2_O and 0.358 Co(NO_3_)_2_·6H_2_O was dissolved in 50 mL deionized water (DI water) and was defined as solution A. The mIM of 3.232 g and 5.49 mL TEA was stirred in 50 mL DI water until dissolved, which was defined as B solution. Solution A was added to solution B, and stirred for 10 min. At this time, the liquid quickly changed from transparent to purple. The mixed solution was separated through centrifugation, the supernatant was decanted, and then the sample was collected and dried in the air in a 60 °C oven for 24 h. Ratio samples of 0.5:0.5:16:4 and 0.5:0.5:4:16 Zn:Co:ligand:TEA were synthesized by half or twice the masses of ligand and TEA, respectively.

### 2.2. Synthesis of ZIF-8, ZIF-67, and ZIF-8-67 Hybrid Membrane

Porous α-alumina (α-Al_2_O_3_) (Fraunhofer IKTS) with a diameter around 30 mm and thickness around 1 mm was selected as the substrate for the membrane growth. The normal pore size is around 160 nm and the porosity is about 35%. The seeding solution was prepared using 5% polyethyleneimine (PEI, Aldrich), 4% sodium bicarbonate (98%, Aldrich), and 30 mL deionized water, and then the pre-synthesized ZIF-8, ZIF-67, and ZIF-8-67 nanocrystals were dispersed in the solution. The seeds were uniformly coated on the aluminum oxide substrate by a spin coating method. After spin coating, the discs were immediately dried in the oven at 60 °C for 24 h. The pretreated support was placed in a microwave autoclave with the secondary growth solutions of 1 mmol ZnCl_2_ (98%, Showa), 1 mmol Co(NO_3_)_2_·6H_2_O (98%, Showa), 4 mmol 2-methylimidazole (mIM, 97%, Acros), and 2.7 mmol sodium formate (SF, 99.5% Aldrich), which were dissolved in methanol (99%, Aldrich) at 100 °C for 4 h.

### 2.3. Characterization

X-ray diffraction (XRD) patterns were collected by recording 2*θ* values ranging from 5° to 50° using a Bruker D8 Advance X-ray diffractometer. The samples (ZIF-8, ZIF-67, and ZIF-8-67) were characterized by sorptometry to determine the surface area via nitrogen adsorption at −196 °C (77 K) in a Micromeritics ASAP 2010 system. Prior to measurement, the samples were activated in situ by holding at 150 °C under a vacuum (0.13 Pa) for 2 h. However, the surface area data were the average of three measurements taken for each sample. Scanning electron microscopy (SEM) images were obtained from a JEOL JSM-6700F instrument operating at 15 kV and a current of 10 µA. The particle size distribution was evaluated using DI water as a dispersion medium at room temperature, and the samples were measured through dynamic light scattering (DLS) by a Photal ELS-8000 (OTSUKA Electronics). Thermal gravimetric analysis (TGA) measurements were recorded on a TA Instrument Q500, which was used to analyze thermal properties from 25 to 800 °C at a rate of 5 °C min^−1^ under nitrogen gas. Ultraviolet–visible spectroscopy (UV–Vis) was carried out in a Cary 300 instrument with a diffuse reflectance accessory in the wavelength range of 200–1000 nm. Gas permeation analyzer (GPA) data were collected on a Yanaco GTR-11MH, and the apparatus was employed to measure the permeability of the hybrid metal ZIF-8-67 membranes for pure H_2_, CO_2_, and N_2_. Tests were carried out at different temperature and pressure conditions. The gas permeability was determined by the following equation:(1)$$P=\frac{q}{\left(P_{1}-P_{2}\right)\cdot A\cdot t}$$
where *P* is the gas permeability (cm^3^ (STP)/cm^2^ s cmHg), *q*/*t* is the volumetric flow rate of gas permeation (cm^3^ (STP)/s), *P*_1_ and *P*_2_ are the pressures (cmHg), and *A* is the effective membrane area (cm^2^). All permeation measurements were repeated on five different samples, at least three times for each sample.

## 3. Results and Discussion

### 3.1. Characteristics of Nano ZIF-8-67 Seeds

The crystal identities of the ZIF-8, ZIF-67, and ZIF-8-67 (Zn:Co = 0.5:0.5) with metal:ligand:TEA ratios of 1:16:16, which were determined by X-ray diffraction (XRD), are shown in Figure 1a. The characteristic diffraction peaks of the ZIF-8 occur at 2*θ* = 7.43, 10.46, and 12.83, whereas those of the ZIF-67 occur at 2*θ* = 7.53, 10.56, and 12.91, suggesting a sodalite structure. The above results show that the characteristic diffraction peaks of the ZIF-67 are slightly higher than those of the ZIF-8. However, the peaks of both ZIF-67 and ZIF-8-67 are shifted to the right relative to those of ZIF-8, which can be attributed to the Co^2+^ ions being smaller than the Zn^2+^ ions. These results qualitatively support the incorporation of Co^2+^ ions into the framework [21]. As shown in Figure 1b, N_2_ adsorption/desorption isotherms of ZIF-8, ZIF-67, and ZIF-8-67 reveal a reversible type I isotherm, characteristic of microporous materials. Table 1 shows that the BET (Brunauer–Emmett–Teller) surface area of hybrid metal ZIF-8-67 (1260.40 m^2^ g^−1^) is much larger than those of both ZIF-8 (1048.33 m^2^ g^−1^) and ZIF-67 (1067.82 m^2^ g^−1^). The result can be attributed to the Co^2+^ ions being smaller than the Zn^2+^ and Co^2+^ ions in the structure of ZIF-8, which creates a larger specific surface area. For this study, a large specific surface area was necessary, as this results in effectively achieving H_2_/CO_2_ gas separation; due to the resultant crystallization of thin film materials, this is one of the most important properties. The thin film was therefore designed to adapt the secondary growth method when the first layer was seeding, during which the powder with a larger specific surface area was exceptionally conducive to the growth of seeds. This was beneficial to the crystallization of the thin film structure.

From the analysis of the SEM images, the ZIF-8 particles were found to have an average diameter of 119 ± 20 nm (Figure 2), and the ZIF-67 particles to have an average diameter of 98 ± 25 nm (Figure 2). The morphology of the ZIF-8 particles was a polyhedral shape. The ZIF-67 particles resembled the ZIF-8 grown in methanol for 1 h, with a ball shape but with non-uniform size distribution. Similarly, the material ZIF-8-67 displayed a polyhedral morphology. This result may be related to the strength of the crystals, which is consistent with the XRD pattern. Typical TGA curves for all three samples are shown in Figure 3, and they all display three weight-loss stages. The first weight-loss stage between room temperature and 200 °C occurs due to occluded solvent and water. The second weight-loss stage of ZIF-67 starts at 200 °C and decreases at 475 °C, whereas ZIF-8-67 and ZIF-8 start to decrease at higher temperatures, up to 520 and 575 °C, respectively. In the third weight-loss stage, the decomposition of organic linkers in the framework caused theatric weight loss. TGA analysis shows that ZIF-8-67 is not as stable as ZIF-8, but is more stable than ZIF-67.

### 3.2. Characterization of ZIF-8-67 Membrane

#### 3.2.1. Membrane Characterization and Effect of Sodium Formate

Figure 4a shows the X-ray diffraction patterns of the ZIF-8-67 membrane with different ratios of (Co^2+^)/(Zn^2+^) + (Co^2+^), and illustrates that ZIF-8 was the only crystalline phase present in all samples, with characteristic diffraction peaks around 2*θ* = 7, 10, and 13, which correspond to the basal planes of (110), (200), and (211), respectively. These patterns of the membranes ZIF-8, Co26-ZIF-8, Co44-ZIF-8, and Co61-ZIF-8 indicate a well-developed crystalline structure that was indexed as the face-centered hexagonal unit cell. The XRD patterns of the Co26-ZIF-8 membrane exhibited a sharply increased relative intensity of the (200) reflection, in relation to other sample reflections, which indicated an optimal crystal orientation of the (100) planes parallel. 

As the Co content increased from Co26 to Co61, the relative intensity of (200) reflection weakened, indicating that the additional Co content affected the adjustment of the deprotonation and nucleation rates. However, this indicates that although the Co^2+^ ion is smaller than Zn^2+^ ion, only a limited amount of Co can be associated with structural development into a stoichiometric ZIF as the Co is incorporated into the ZIF-8 framework. Additionally, Co, such as Co61-ZIF-8, would affect the oriented growth. As shown in Figure 4b, the XRD patterns for all Co44-ZIF-8 membranes exhibit a highly crystalline pure-phase ZIF-8-67 structure, but the ZIF-8-67 with SF of 5.4 mmol exhibits a more vigorous intensity of the (200) reflection. 

According to a previous study [22], ZIF crystals are nucleated on support surfaces, and sodium formate can act as a deprotonator, resulting in uniform and stable growth of crystals in all directions, thus producing a continuous well-symbiotic crystal. When the SF was increased from 2.7 to 5.4, it likely caused ZIF-8-67 to become fully deprotonated, due to the increase in pH value, and the solution remained transparent under atmospheric conditions. However, when the 10.8 mmol SF was used, the solution became blurred within 10 min. These observations clearly indicated that more deprotonated mIM ions could be coordinated with metal ions as pH concentration increased. As a result, the nucleation rate increased, decreasing crystal growth on the membrane surface. Consequently, the ZIF-8-67 to which 10.8 mmol SF was added displayed a lower (200) reflection.

SEM images of the synthesized membranes with SF content from 2.7 to 10.8 mmol are shown in Figure 5. The top views of the morphology of the grains synthesized from 5.4 mmol are more uniform than those synthesized from 2.7 mmol of SF, with grain size also increasing to 10 µm with increasing SF content (Figure 5a,c), indicating that the membrane was fully deprotonated in the 5.4 mmol membrane. The cross-sectional images in Figure 5b,d reveal that the thickness of the membranes synthesized with 2.7 and 5.4 mmol significantly increased from 5 µm to 22 µm, and displayed a compact structure with oriented intergrown grains. They also provide evidence that when there is enough SF, the mIM can be deprotonated, but still grow on the membrane surface to fabricate a thicker membrane of ~22 µm. 

In contrast, when increasing the SF content to 10.8 mmol, a non-uniform grain morphology was observed, and the crystal size decreased significantly from 10 to around 5 µm, as shown in Figure 5e. The cross-sectional view of the membrane demonstrates a reduced thickness from 22 to 10 µm. This result suggests that the nucleation and crystal growth rates that control thickness are strongly influenced by the concentration of SF in the second growth solution.

#### 3.2.2. Effect of Ligand

Figure 6a shows the X-ray diffraction patterns of the hybrid metal ZIF-8-67 membrane with different ratios of (Co^2+^)/(Zn^2+^) + (Co^2+^) synthesized from 2, 4, 8, and 16 mmol 2-methylimidazole (mIM). The results showed that the whole crystal phase was the same as that of the ZIF-8 membrane, and no other crystal phase was derived. We noted that with the addition of less than 4 mmol of mIM, almost all characteristic diffraction peaks of ZIF-8-67 disappeared, and the synthesized materials became amorphous. Conversely, the membrane with more than 2 mmol of mIM added had diffraction patterns identical to those of the ZIF-8 membrane, and a relatively strong intensity of the (200) reflection appeared in the Co44-ZIF-8 membrane with 4 mmol mIM, indicating that the Co44-ZIF-8 crystal surface was fully deprotonated in the 4 mmol mIM. However, a further increase in the mIM proportion caused a decreased intensity of the (200) reflection, due to the phenomenon that more mIM at a fixed SF molar content causes part deprotonation, resulting in the nonuniform growth of the Co44-ZIF-8 membrane. 

To further confirm the effect of Co^2+^ ion coordination in the mIM on direct synthesis of the membrane in the hybrid metal system, UV–Vis analysis was performed, and the results are shown in Figure 6b. The UV–Vis spectra of the Co44-ZIF-8 secondary growth solution, with different molar ratio ligands and cobalt (II) nitrate hexahydrate dissolved in methanol, indicate the presence of Co^2+^ ions, as well as their tetrahedral coordination mode in the Co44-ZIF-8 structures. We also noted that in comparison to the UV–Vis spectra, the solutions in Figure 6c show colors from pink, to blue, to violet, corresponding to 2, 4, and 8 mmol ligands containing Co44-ZIF-8 secondary growth solution, which is likely due to a Co^2+^ coordination shift from octahedral to tetrahedral. In other words, the tetrahedral coordinated Co^2+^ sites can be promoted by adding 4 mmol ligands to the solution. In a ligand of over 4 mmol, the Co44-ZIF-8 secondary growth solution does not further change its color.

The SEM top-view microstructures of the hybrid membranes synthesized with different molar concentrations of mIM are presented in Figure 7. The membrane grown from the solution containing 4 mmol mIM displayed a continuous and high impact structure, and most grains were highly symbiotic with each other on the α-Al_2_O_3_ substrate. As mIM increased to 8 mmol, a weak intergrown membrane with rhombic dodecahedral shape was observed. With a further increase to 16 mmol mIM, the Co44-ZIF-8 membrane displayed random and discontinuous crystal growth on the surface. This caused more mIM to act as a capping agent, which terminated crystal formation.

#### 3.2.3. Effect of Microwave Solvothermal Temperatures

To better understand the formation of the Co44-ZIF-8 membrane under different reaction temperatures, within microwave solvothermal synthesis, Figure 8 shows the XRD patterns of the Co44-ZIF-8 membrane synthesized at different temperatures. When the reaction temperature is lower than 80 °C, there are no ZIT-8-67 peaks. Such a phenomenon means that the crystal of Co44-ZIF-8 can notfound and form nucleation at a relatively low temperature. A thin amorphous layer appears on the alumina surface, indicating it is not suitable for crystal growth on the support. Once the reaction temperature increases to 100 °C, crystals characteristic of ZIF-8-67 are continuous and adjacent to one another, corresponding to a high-intensity XRD pattern. However, with the temperature further increasing to 150 °C, a discontinuous layer along the bare support is formed, which is consistent with low peak intensities in the XRD pattern. To further understand the effect of the reaction temperature on the nucleation and growth of Co44-ZIF-8, SEM micrographs were recorded.

### 3.3. Gas Permeation Performance of Co44-ZIF-8 Membranes

The permeation performances of the ZIF-8 and Co44-ZIF-8 membranes were measured using a gas permeation setup. For comparison, we controlled the membrane thickness and crystal orientation of both Co44-ZIF-8 and ZIF-8 membranes so they were identical, with no visible cracks, or other defects observed. Figure 9 shows the gas separation performance of the Co44-ZIF-8 hybrid membrane in terms of different transmembrane pressure drops, i.e., the volumetric flow rates of the single gases (H_2_, N_2_, and CO_2_) through the membrane at room temperature. The single-component gas permeance decreased with increased molecular weights in the order H_2_ > N_2_ > CO_2_. The dominance of gas permeance in diffusion control, but not the molecular sieve control, suggests that the Co44-ZIF-8 on the membrane surface has more interaction with CO_2_ than N_2_ [23,24], which has a significant influence on the permeability through the Co44-ZIF-8 membrane. This trend is more obvious in the lower pressure drop. Both CO_2_ and N_2_ permeance slightly increases with increasing the pressure drop, but H_2_ permeance decreases with the pressure drop. A further increase in the pressure drop of H_2_ could enhance the permeation flux, but the increase would not be sufficient to compensate for the higher pressure difference, resulting in decreased permeability. Figure 9b shows that the gas permeance selectivity (α) decreased in αH_2_/CO_2_ from 12.4 to 9.0, and in αH2/N2 from 3.6 to 2.8, with increasing feed gas pressure.

The single component permeability of the gas kinetic diameter test is illustrated in Figure 10. This ZIF-8 membrane has higher H_2_ and CO_2_ gas permeance but lower H_2_/CO_2_ selectivity than ZIF-8-67. Notably, our synthesis-oriented ZIF-8 selectivity is lower than in the previous reports, which also noted orientation [14,15]. This may be related to the diameter size and character of the α-Al_2_O_3_ disc support. In this study, a larger diameter (30 mm) α-Al_2_O_3_ disc was used, which would have a higher chance of producing more defects on the surface, making the membrane density lower. Furthermore, as our seeding ZIF-8 particle size (~100 nm) was larger than in the literature (< 50 nm), it is also possible that this caused poor stacking in the coating step. A comparison with both Co44-ZIF-8 and ZIF-8 membranes for gas selectivity can be found in Figure 10b. The membrane is composed of Co44-ZIF-8 with a H_2_/CO_2_ selectivity of nine, which remarkably exceeds the selectivity of the ZIF-8 membranes. The main reason would be the slightly larger pore volume of the Co44-ZIF-8 membrane (0.616 cm^3^ g^−1^) compared with the ZIF-8 membrane (0.485 cm^3^ g^−1^). The Co^2+^ ions incorporated into the ZIF structure are beneficial for the Co44-ZIF-8 on the membrane surface, and result in more CO_2_ being adsorbed and diffused into a pore channel.

We can compare this result with those of Jang et al. [25], who reported a maximum H_2_/CO_2_ selectivity of ~7.5 ± 0.2 for the membrane, ZIF-8_α. Although the ZIF-8_γα membrane showed a marked H_2_ performance, with a maximum H_2_/CO_2_ separation factor of ~9.9 ± 1.2, the test temperature was 250 °C.

Previous works demonstrated that MOF membranes’ performance for gas separation might be significantly affected by the actual ambient temperature applied. Consequently, the ZIF-8 and Co44-ZIF-8 membranes was also investigated at different temperatures. Figure 11 shows that when the feed gas is a single component, the permeance values slightly decrease with the increase of the measured temperature. In contrast, CO_2_ shows a different behavior compared to the ZIF-8 membrane: a decrease in gas permeance when the temperature reaches 330 K, then an increase in gas permeance with further increasing temperature. As described by Zhao et al. [26], gas permeability is mainly dependent on temperature, and it can increase or decrease with increasing temperature, which depends entirely on the relative values of activation energy for diffusion, and exothermic adsorption. The probable cause for the CO_2_ behavior may be the change in the two related properties with the increase in temperature. For the above experimental results, this can be attributed to the Co44-ZIF-8 membrane that is conducive to enhancing the strong adsorption of CO_2_ molecules. However, a large quadrupole moment and polarizability facilitated the strong CO_2_ adsorption [27]. In addition, it must be considered that the rate of diffusion in the channels or pores of the membrane microstructure also has a significant influence on permeability [26,27]. In addition, the sucking behavior of CO_2_ induces the ideal selectivity to exhibit maximum value as a function of temperature, and this information is beneficial for getting the best-operating conditions for such a membrane. This can be explained by the Co^2+^ ions in the hybrid metal membrane interacting strongly with CO_2_, which influences the gas permeance and leads to better H_2_/CO_2_ selectivity.

To prove the stronger interaction between the CO_2_ and the ZIF-8-67, the powder was collected after secondary growth using a microwave hydrothermal method. The adsorption and desorption of CO_2_ for ZIF-8 and Co44-ZIF-8 displayed hysteresis loops at 25 °C, and these are shown in Figure 12. The maximum CO_2_ adsorptions of ZIF-8 and Co44-ZIF-8 at 1 atm were 0.88 and 1.07 mmol g^−1^, respectively. Co44-ZIF-8 contributed more to the CO_2_ uptake compared to ZIF-8. The interaction of CO_2_ molecules with the Co44-ZIF-8 material can be attributed to the fact that the CO_2_ molecule has a significant quadrupole moment to induce specific interactions with the adsorbents. Therefore, the integrated Co44-ZIF-8 membrane is thought to have considerable potential in the practical application of hydrogen and carbon dioxide gas separation.

## 4. Conclusions

As a new crystalline, microporous material, metal-organic frameworks have received much interest from membranologists, particularly those engaged in microporous membrane research. For such frameworks to be used as membrane materials, careful consideration should be given to including thermal, hydrothermal, and chemical stability. We synthesized hybrid metal nano ZIF-8-67 crystals to fabricate a novel hybrid metal ZIF-8-67 membrane with the secondary growth method, and using a microwave hydrothermal method. It is the first study to use hybrid metal as a gas separation membrane. In the first part of the study, we used a proficient method for the preparation of ZIF-8-67 crystals with adjusted particle sizes, and in the second part, we successfully developed the hybrid metal ZIF-8-67 membrane on a porous α-alumina substrate, using secondary growth synthesis. The membrane not only has a preferred (200) reflection but also exhibits outstanding thermal stability and excellent quality. Single-gas permeation of the hybrid metal ZIF-8-67 membrane demonstrated smaller permeation than the ZIF-8 membrane, but the selectivity was better than the ZIF-8 membrane. The framework with hybrid metal, which contains Co^2+^ ions, had strong interactions between CO_2_ and the hybrid metal ZIF-8-67 membrane, which were confirmed to adsorb and desorb CO_2_ at 298 K. Our study provides a novel idea for the fabrication of hybrid metal ZIF-8-67 membranes, as well as an improvement in H_2_/CO_2_ gas separation performance. The results of our work will help ZIF membrane research, and can hopefully be further applied in industry in the future.

## Figures and Tables

**Figure 1 materials-13-05009-f001:**
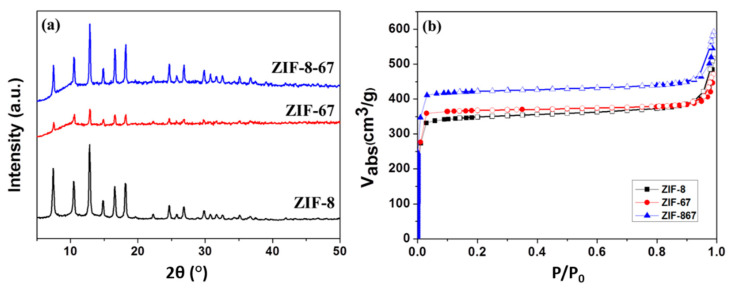
(**a**) XRD patterns of ZIF-8, ZIF-67, and ZIF-8-67 samples; (**b**) comparison of N_2_ adsorption/ desorption isotherms of ZIF-8, ZIF-67, and ZIF-8-67 samples.

**Figure 2 materials-13-05009-f002:**
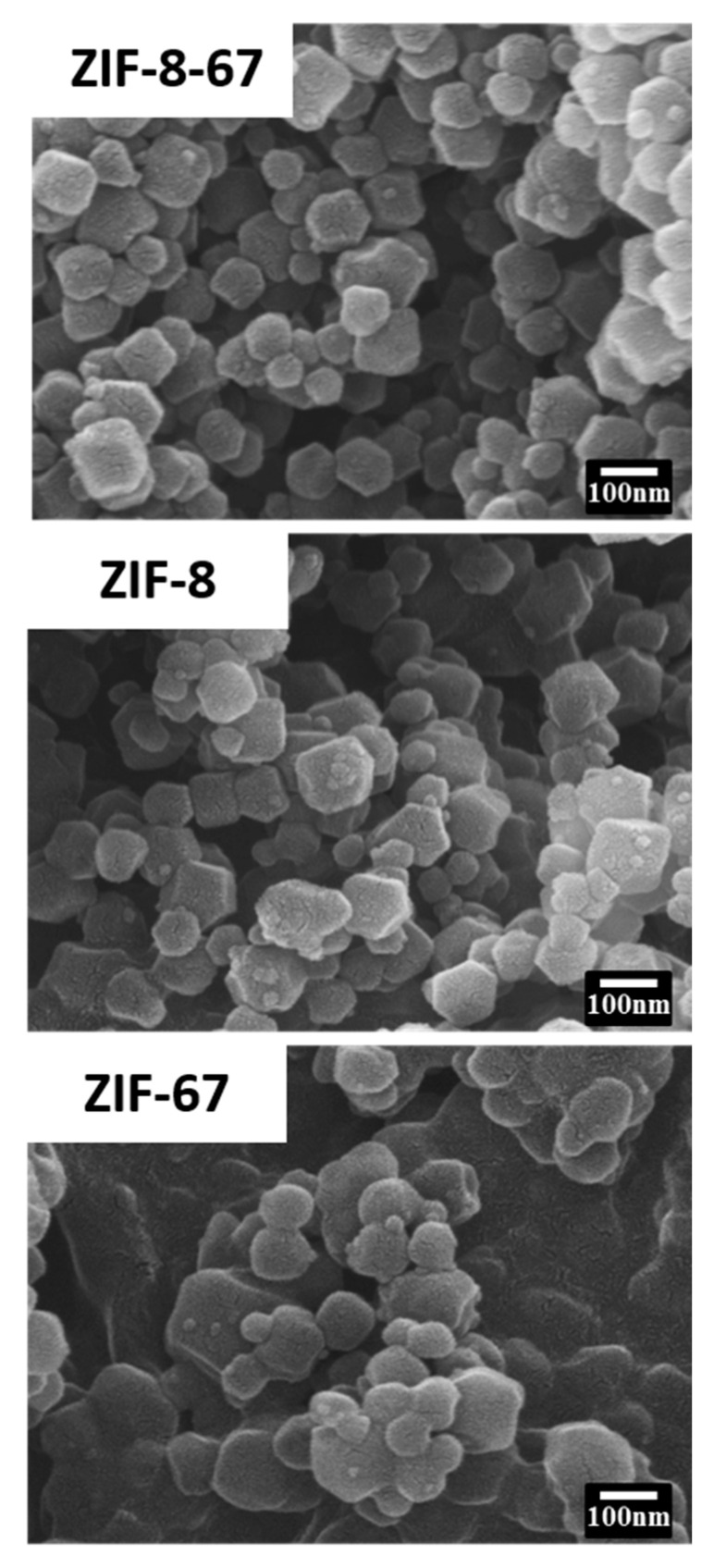
SEM images of ZIF-8, ZIF-67, and ZIF-8-67.

**Figure 3 materials-13-05009-f003:**
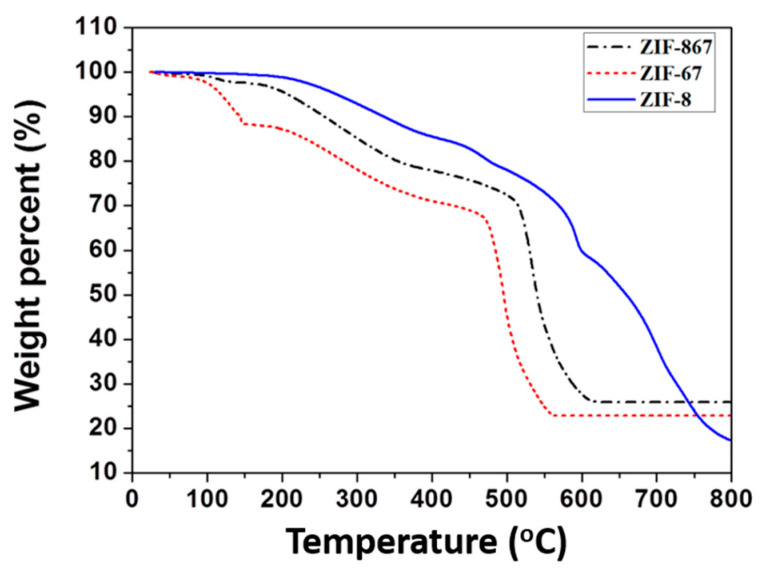
TGA curves of metal: ligand: TEA ratios of 1:16:16 for ZIF-8, ZIF-67, and ZIF-8-67 (Zn:Co = 0.5:0.5).

**Figure 4 materials-13-05009-f004:**
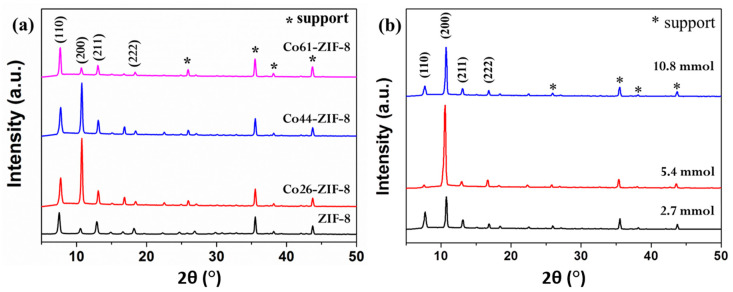
XRD patterns of ZIF-8-67 membrane: (**a**) with different Co content and (**b**) structural evolution of the Co44-ZIF-8 membrane with different SF concentrations (2.7, 5.4, and 10.8 mmol at 100 °C for 2 h).

**Figure 5 materials-13-05009-f005:**
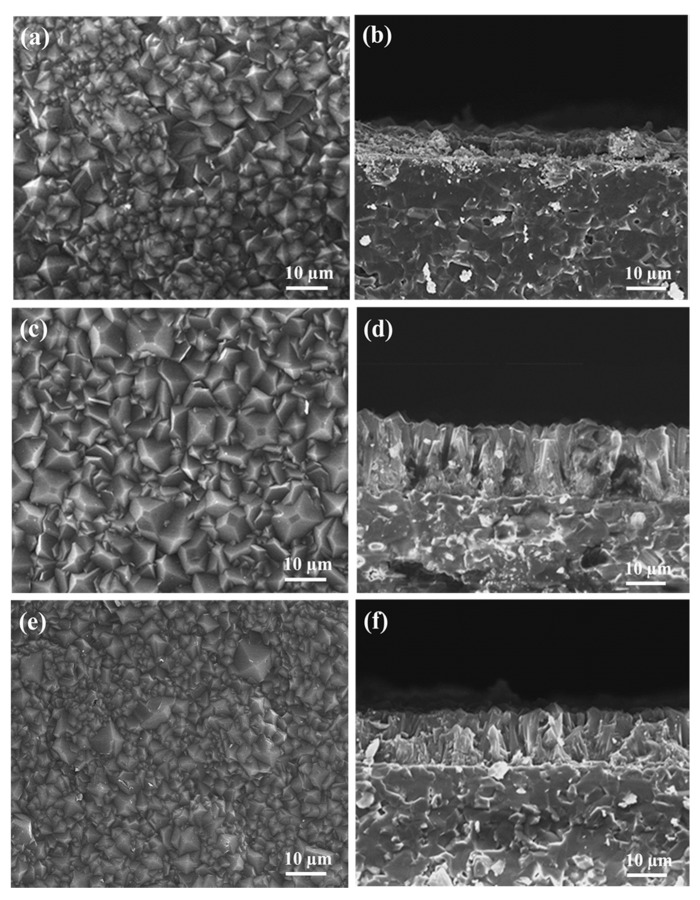
Top-view and cross-section SEM images of the Co44-ZIF-8 membrane with different sodium formate concentrations: 2.7 mmol (**a**,**b**), 5.4 mmol (**c**,**d**), and 10.8 mmol (**e**,**f**) at 100 °C for 2 h.

**Figure 6 materials-13-05009-f006:**
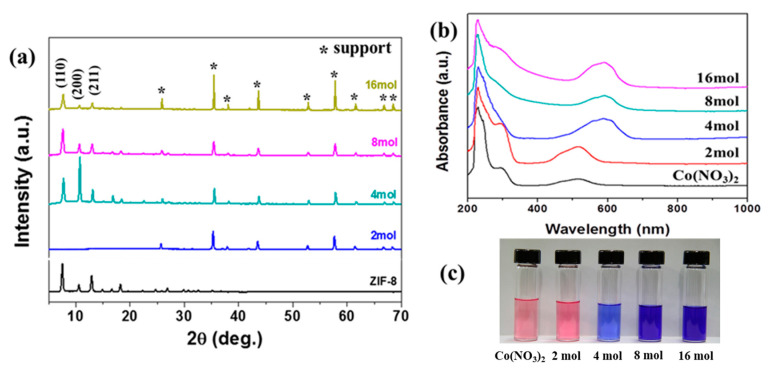
(**a**) XRD patterns for Co44-ZIF-8 membrane with different mIM concentrations: 2, 4, 8, and 16 mmol at 100 °C for 2 h, (**b**) UV–Vis spectra, and (**c**) color changes of synthetic solution.

**Figure 7 materials-13-05009-f007:**
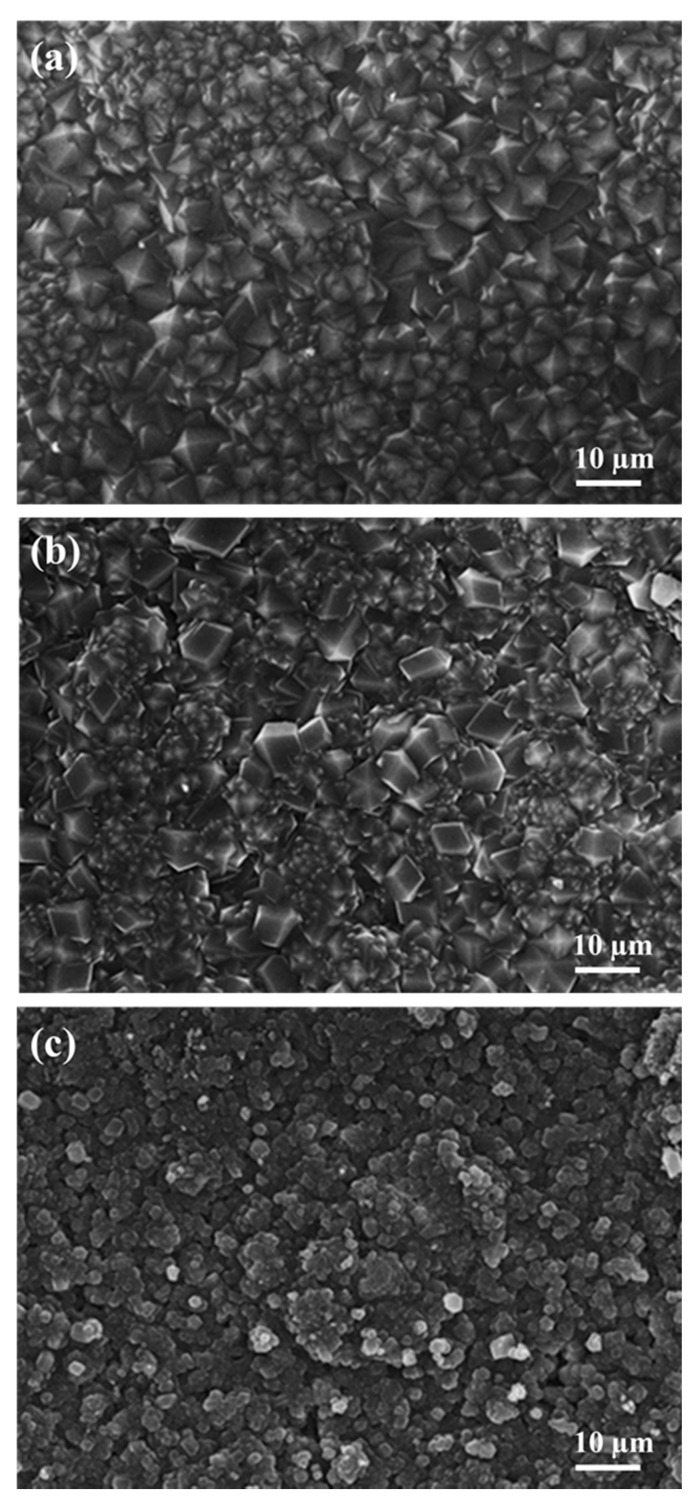
SEM top-view image of Co44-ZIF-8 membrane with different amounts of mIM: (**a**) 4 mmol, (**b**) 8 mmol, and (**c**) 16 mmol.

**Figure 8 materials-13-05009-f008:**
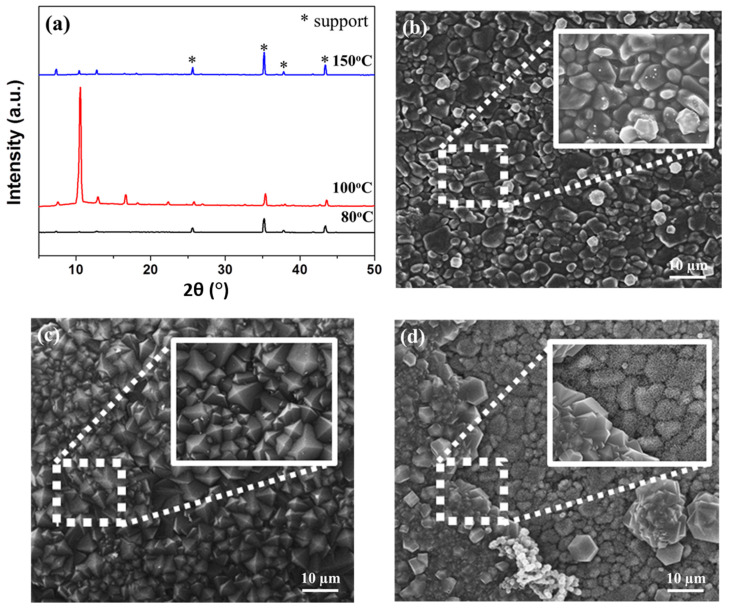
(**a**) XRD patterns and SEM images of Co44-ZIF-8 membrane prepared with different reaction temperatures: (**b**) 80, (**c**) 100, and (**d**) 150 °C.

**Figure 9 materials-13-05009-f009:**
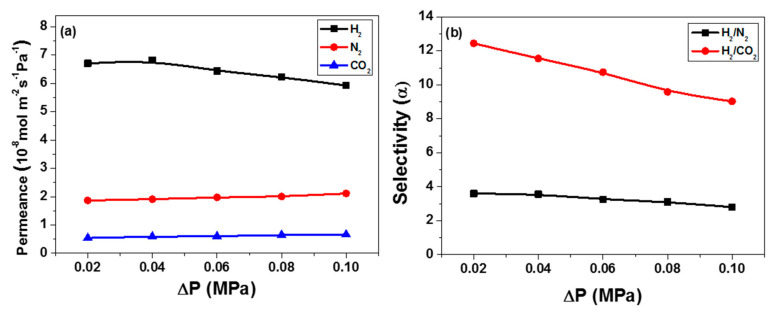
Gas permeance through the Co44-ZIF-8 membrane at different transmembrane pressure drops for (**a**) single gas permeance and (**b**) selectivity.

**Figure 10 materials-13-05009-f010:**
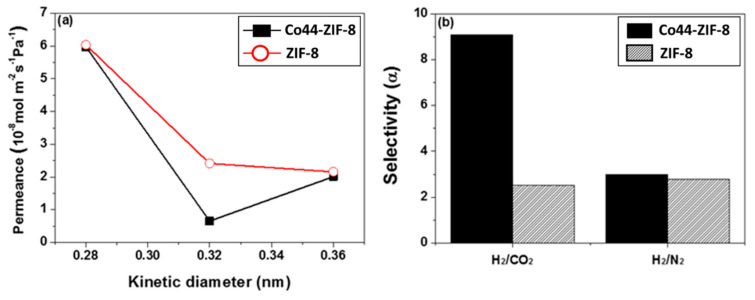
Permeances of single gases for ZIF-8 and Co44-ZIF-8 (**a**) as a function of the kinetic diameter (H_2_: 0.28, CO_2_: 0.32, and N_2_: 0.36 nm) at 25 °C and 0.1 MPa, and (**b**) the separation factors for H_2_/CO_2_ and H_2_/N_2_.

**Figure 11 materials-13-05009-f011:**
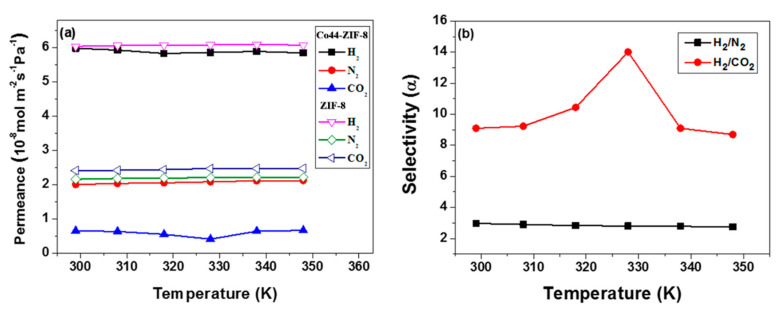
Single component gas permeation of H_2_, N_2_, and CO_2_ results for (**a**) ZIF-8 and Co44-ZIF-8 membranes, and (**b**) ideal selectivities as functions of temperature.

**Figure 12 materials-13-05009-f012:**
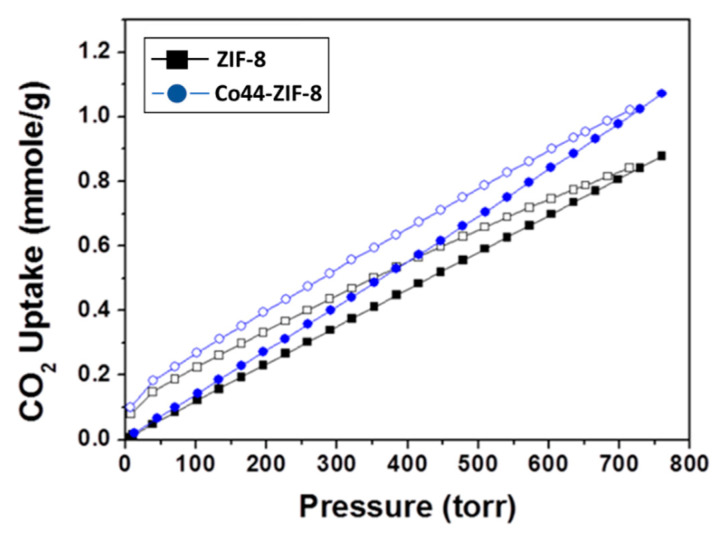
CO_2_ adsorption–desorption of ZIF-8 and Co44-ZIF-8.

**Table 1 materials-13-05009-t001:** Zeolitic imidazolate framework (ZIF) porosimetry data.

Material	BET Surface Area (m^2^ g^−1^)	Micropore Volume (cm^3^ g^−1^)
ZIF-8	1048.33	0.485
ZIF-67	1067.82	0.542
ZIF-8-67	1260.40	0.616

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
