# Peer review of "Synthesis and Characterization of Hybrid Metal Zeolitic Imidazolate Framework Membrane for Efficient H2/CO2 Gas Separation"

_materials, 2020, doi:10.3390/ma13215009_

Round 1
Reviewer 1 Report
The authors added significant work to improve quality of the paper. However there are few things to solve:
1) The language of the manuscript and abstract must be improved, because of some misunderstandings and not correct grammar.
2) The results of the particle size measurements should be added to the paper with explanation of the measurement in the "Methods" section.
Author Response
Response to Reviewer 1 Comments
Point 1: The language of the manuscript and abstract must be improved, because of some misunderstandings and not correct grammar.
Response 1: We really appreciate your kind suggestion and careful revision. We have tried to fix these grammar errors and modify these sentences to make it clear. Also, we will ask MDPI for making a thorough English revision of this manuscript.
Point 2: The results of the particle size measurements should be added to the paper with explanation of the measurement in the "Methods" section.
Response 2: Thanks for kind suggestion.
The particle size distribution was evaluated using DI water as a dispersion medium at room temperature and the samples was measured through dynamic light scattering (DLS) by a Photal ELS-8000 (OTSUKA Electronics).
We have added particle size measurements in the experimental methods section. (Line 135 ~ Line 138, Page 3)

Reviewer 2 Report
The authors have carried out an original work. The manuscript includes relevant and sufficient information on its topic.
Author Response
Response to Reviewer 2 Comments
Point 1: The authors have carried out an original work. The manuscript includes relevant and sufficient information on its topic.
Response 1: Thanks for the positive comments.

Reviewer 3 Report
The manuscript "A Novel Zeolitic Imidazolate Frameworks Membrane by Mixed-Metal Synthesis for Efficient H2/CO2 Gas separation" has been reviewed.
It's the modified resubmission of the manuscript 898738 with the same title.
The manuscript has been deeply revised according to the reviewer's suggestions. All the issues have been addressed and it can be accepted for publication in the present form.
Author Response
Response to Reviewer 3 Comments
Point 1: The manuscript "A Novel Zeolitic Imidazolate Frameworks Membrane by Mixed-Metal Synthesis for Efficient H2/CO2 Gas separation" has been reviewed. It's the modified resubmission of the manuscript 898738 with the same title. The manuscript has been deeply revised according to the reviewer's suggestions. All the issues have been addressed and it can be accepted for publication in the present form.
Response 1: We really appreciate your positive comments.

This manuscript is a resubmission of an earlier submission. The following is a list of the peer review reports and author responses from that submission.
Round 1
Reviewer 1 Report
- Introduction is very chaotic. Despite the overview of the published papers in this area introduction lacks clear explanation what are the differences between ZIF-8, ZIF-67 ant ZIF-8-67 and the membranes with inserted Co. From lack of the explanation is not clear what the idea of current work is 7and what its novelity.
- In the text sometimes is misunderstandings if the talk is about the ZIF-8-67 membrane or ZIF-8 membrane or c044-ZIF-8 membrane. It is not clear which of these membranes is the aim of the research. In the experimental section preparation of the ZIF-8-67 is explained, but in the results section we can observe also ZIF-8 and ZIF-67 membranes,so it is not clear is this a mistake or not.
- In the XRD measurements presented in figure 1 it is not any clearly visible difference in the spectra of different membranes, so the picture must be reformated and more explanation must be added on the XRD analysis of these membranes.
- From the SEM pictures presented in Figure 2, the average particle size is calculated but its not clear how it was calculated. The size overlaps with the range of error, so it is not clear is there any difference in size for the different membranes.
- It is not clear what BET and TGA means as there is no explanation in the paper.
- As the main aim of the work is not clear, so it is also not clear that the results proves the aim of the work.
- In the results it is not clear how the different measurements are related as in each chapter different membrane is the thing of interest.
Reviewer 2 Report
The authors have shown an interesting work. The points to be revised are mentioned below:
1) In line 56, the reference number notation should be corrected as [8].
2) The paragraph between lines 60-72 does not include references. Including citations from relevant literature which supports the claims made in the paragraph is required to support the content presented.
3) The last paragraph of the introduction should be divided into two. In the revised version, the last paragraph of introduction section should be the part in which the novelty of the manuscript is explained.
4) In lines 292-293, the related references should be mentioned.
5)The conclusion section should be consisting of the points related to the main outcomes of the work presented and future perspectives. So the part till the phrase that starts with "In the current study, we..." should be removed in order to obtain a more concise structure
Reviewer 3 Report
The manuscript "A Novel Zeolitic Imidazolate Frameworks Membrane by Mixed-Metal Synthesis for Efficient H2/CO2 Gas separation" has been reviewed.
The manuscript may be accepted after the following minor revisions:
1) Acronyms (XRD, FTIR) should be introduced;
2) Line 56: ref. [8] not [18]
3) Line 241: 80 °C
4) XRD and UV-VIS spectra: please increase legibility.
5) Line 363: ref. 9 is not reported.
6) More in general references may be enriched by more recent articles.
Reviewer 4 Report
THe introduction lacks any references around mixed-metal ZIFs and does not justify claims such as Line 67-68 "the second
67 metal contributes nothing to the framework structure's construction but resides within it as an
68 attachment"
There are many references to chose from, some which indicate the an additional metal does not "reside within" e.g. https://pubs.acs.org/doi/abs/10.1021/acs.inorgchem.6b00814
Lines 69-71 contain the claim: "The cobalt was selected because of its excellent
70 interaction property with carbon dioxide, making the membrane useful for gas separation
71 application."
Likewise, this claim should be referenced (if true; I am unaware of any sicentific research suggesing cobaly has increased affinity for CO2 compared with zinc; espeically when zinc containing enzymes are used to transport CO2 in biological systems).
In the subsequent paragraph Lines 80-81 are included with a reference to work that suggests physical changes induced by inclusion of cobalt are responsible for differences in selectivities and permeances.
Results and Discussion:
The changes in XRD patterns are minor and insuffiecient to draw conclusions about the inclusion of cobalt into the ZIF structure. I recommend additional techniques be used. E.g. Energy Dispersive X-Ray Analysis (EDX) to determine the elemental composition.
The surface area reuslts should include error bars showing three separate measurements on separate batches of materials. Single data points hide variation in batches of materials and result in erroneous assertions.
The TGA data is the most convincing that a ZIF-8-67 structure has been produced; the authors state this data is 'representative' and should include replicates in the supporting informaiton
Line 269/270: "However, the membrane maintains selectivity at 1 bar, which is close to the case in industrial
270 condition." THis is untrue for the majority of gas separation processes.
The selectivities are low and compared with ZIF-8 data collected by these authors and so low it suggests a defective membrane. Literature data on ZIF-8 membranes should be used as a comparison.
Lines 320-322 are not justified in any way. E.g. a process model showing that a membrane of thse permeabilities and selectivities would be useful for any application.
Lines 340-342: No reasoning or supporting evidence is given why these resutls will either advance scientific understanding or be applied industrially.